# The Crazy Biology

**DOI:** 10.3390/genes13101769

**Published:** 2022-09-30

**Authors:** Philippe Monget

**Affiliations:** Physiologie de la Reproduction et des Comportements, Centre Val de Loire—UMR INRAE, CNRS, IFCE, Université de Tours, 37380 Nouzilly, France; philippe.monget@inrae.fr

**Keywords:** species-specific gene duplication, cell count, crossroad between biological functions

## Abstract

Since the end of the 1980s and the advent of molecular biology, then the beginning of the 2000s with the sequencing of whole genomes, modern tools have never ceased to amaze us and provide answers to questions that we didn’t even dare ask ourselves before: Why do elephants have fewer cancers than humans? Why do humans have such big brains? How does a eukaryotic cell recognize a “foreign” DNA sequence? Are there molecular crossroads of incompatible functions? Can cells count each other? These fascinating questions have made biology in recent years almost crazy.

## 1. Biology Is an Amazing World

Biology is a fascinating world, where many researchers catch stars in their eyes when they simply look at a histological section or cells in culture, observe the phenotypic consequences of a mutation in an animal or a plant, or decipher the complex genetic regulation responsible for the flowering of plants or for a tissue organization plan. There are also cases where a biological phenomenon is so astounding that it becomes almost unbelievable, even when a hypothesis is able to explain it or a molecular mechanism that underlies this phenomenon has been highlighted. The past thirty years have witnessed such diverse and fascinating sightings, and it is time to start presenting some of them in 2022. This small review aims at presenting some examples of these extraordinary phenomena, in various scales and fields, without pretending to be exhaustive.

### 1.1. Low Incidence of Cancers in Big Animals

At least in mammals, the body size and lifespan are strongly correlated, the lifespan being more than 50 years for African elephants. Assuming that there are much more cells in the body of this species than in human, one would have expected a high incidence of cancer in the African elephant, and yet this is not the case. Two reasons could explain this observation. Firstly, it was recently suggested that somatic mutation rates may be a contributing factor in ageing. Indeed, among different mammalian species exhibiting a variation of around 30-fold in lifespan and around 40,000-fold in body mass (human, mouse, rat, naked mole rat, dog, cattle, tiger, giraffe, etc.), the somatic mutation rate per year exhibits a strong inverse relationship with the species lifespan [1].

A second explanation, not incompatible with the first, would be an increase by duplication of molecular tools in the genome which would make it possible to protect the latter from carcinogenic mutations. For example, the genome of the African elephant contains a massive duplication of p53 retrogenes (Figure 1), which allows this species to have around twenty copies of the corresponding protein to protect the genome [2]. This observation contributes to **Peto’s paradox**, i.e., the lack of correlation between body size and cancer risk [3]. What is certain is that it is not the size of the naked mole rat that is responsible for the very high life expectancy of this species compared to other rodents and the low incidence of cancer in this species [4].

### 1.2. The Patchwork Mutation with Suicide below a Threshold

Homozygous patchwork mice are characterized by hairs that are either totally white or totally pigmented, leading to a ‘salt-and-pepper’ appearance of the coat (on a ‘non-**agouti**’ genetic background). The absence of melanocytes in the follicles of the white hairs is due to premature apoptosis of melanoblast cells at day 18.5 of foetal development as long as the number of apoptotic cells has not exceeded a threshold [5]. Below this threshold, these melanoblast cells commit suicide! One hypothesis could be that the melanoblasts in the hair follicles secrete growth factors, which have a minimum local level that is essential for the survival of these cells (Figure 2).

### 1.3. The Size of Human Brain

The evolution of species can occur, among other things, by **positive selection** of genes which can promote the diversification of coding sequences and thus the diversification of the biological functions of proteins. In 2008, it was shown that positive selection of the abnormal spindle-like microcephaly-associated (ASPM) gene was correlated to the increase in the relative cerebral cortex size in several primates in comparison with other mammals, in particular rodents, their neighbours in the tree of life [6]. Moreover, the evolution can also occur by gene birth, through gene duplication. Very recently (less than 5 **MYA**), segmental duplications of two genes are responsible for the human-specific increase in the size of the cerebral cortex, the gene encoding the Slit-Robo Rho GTPase-activating protein 2 (SRGAP2), and the gene encoding the Rho-type GTPase-activating protein 11A (ARHGAP11A) (Figure 3). Concerning both genes, duplications were partial, producing incomplete proteins that enhance the proliferation of radial glial neuron progenitor cells before their differentiation [7]. Another segmental duplication of the NOTCH2 gene specifically occurred in the human genome, leading to three NOTCH2NL paralogs, which produces proteins that are able to enhance the proliferation of radial glial cells before the differentiation of neurons as well [8]. More strikingly, deletions or further duplications of NOTCH2NL genes are responsible for neurodevelopmental disorders, such as autism spectrum disorder, developmental coordination disorder, intellectual disability, or extreme anxiety and mood disorders. It is as if these neurological human diseases are the price to pay for having a big brain [8].

### 1.4. Loss of Function of SOCS2: More Milk, More Mastitis

Genetic selection of animals and plants can lead to incompatibilities between two biological functions. For example, there are dairy ewes whose genetic selection has resulted in an increase in height at the withers and an increase in milk production but also an increase in the frequency of mastitis, due to an immune reaction following a microbial infection. The positive correlation between milk production and the rate of mastitis is very frequent in ruminants. Growth hormone and interleukin 2 receptors belong to the same family of cytokine receptors. Among the elements involved in the signalling pathways of these receptors are the inhibitory factor cytokine-signalling suppressor protein (SOCS). A mutation has been identified/selected in the **SH2 domain** of the SOCS2 gene in a flock of ewes of the Lacaune breed (whose milk is used to make Roquefort cheese in Aveyron, France). This R96C mutation is responsible for a complete loss of function of the SOCS2 protein, which is responsible for lifting the brake on the action of growth hormone, increasing height at the withers and milk production, and also increasing the frequency of mastitis, a small microbial infection leading to an oversized immune reaction [9]. To my knowledge, this is the first and only example of the identification of a mutation in a single gene which is exactly at the crossroads between two incompatible biological functions. French breeders are in the process of eliminating this mutation, which will reduce milk production and the number of animals suffering with mastitis.

### 1.5. A Chi-Square Analysis of the Codon Use in Mammalian Cells?

When a DNA sequence from a species far from mammals in the tree of life (as the cre recombinase and the **lacZ** prokaryote genes or **GFP**) is injected into a eukaryotic cell, such as an oocyte, this DNA is recognized as a foreign sequence because it contains an abnormally high number of CpG sequences and the preferential use of the genetic code is not the same. For example, the prokaryotic cre contains an undesirably high frequency of CpG dinucleotides (65), and its codon usage is not the same as that for eukaryotes, leading to epigenetic silencing via methylation during early development [10] (Figure 4). This ability to methylate a foreign sequence depends on the genetic background of the mouse [11]. It is as if the recipient cell performed a chi-square analysis of the proportions of codons used in the open reading frame of the foreign sequence and saw that the percentage of codon usage to encode an amino acid was significantly different from its own! The recipient cell does not like it, so it methylates the DNA. Researchers have found a solution by adapting the use of the codons to a ‘humanized’ use, where the level of expression of cre is very significantly increased [12]. This was the case with jellyfish GFP [13] and the lac repressor [11].

## 2. Concluding Remarks—Opening Questions

These few examples illustrate how biology science is an incredible and diverse world that may cause perplexity as well as amazement. Biological complexity still resists and challenges current biotechnologies, even the most effective. We still need to work on them, but overall to expand interdisciplinary collaborations to challenge new and exciting biological concepts. For example, should we systematize the search for species-specific gene duplications and also positive selection that could explain crucial biological phenomena as well as species differences? Additionally, how do the cells quantify signals such as the relative codon usage or the number of neighboring cells? By a stoichiometric assay of tRNA and sensing of growth factors? Finally, how do we systematically identify genes that are at the crossroads of contradictory biological functions? By increasing/facilitating interdisciplinary research more systematically? I hope that young researchers starting their careers will be able to meet these challenges.

## 3. Glossary

**Agouti**—In Mendelian genetics, characterizes the colour of the coat of mammals containing hairs with black bands interspersed with lighter bands. The base colour of the hair is constituted by pheomelanin and the black bands by eumelanin. This type of hair is reminiscent of the coat of the animal called agouti, which has a mottled brown coat.

**GFP**—Green fluorescent protein is a jellyfish protein with the property of emitting a green-colored fluorescence under the action of a luciferase. Its gene can be fused in vitro to the gene encoding a protein whose cellular localization one wishes to study.

**LacZ**—The *E. coli* LacZ gene encodes a protein which produces a blue component once it is cleaved by the β-galactosidase enzyme. This gene is frequently used to visualize the activity of a promoter in vivo in mice or zebrafish.

**MYA**—million years ago.

**Peto’s paradox**—This paradox corresponds to the lack of correlation between cancer risk and body size. Animals with 100 to 1000 times more cells than humans do not exhibit an increased cancer risk, suggesting the existence of mechanisms of protection against cancer.

**Positive selection**—Positive selection of coding genes is an evolutionary mechanism that promotes/maintains nucleotide changes that generate amino acid changes (at the first or second nucleotide of codons).

**SH2** (**S**rc **H**omology **2**) **domain** is a protein domain very well conserved among living organisms, present in the Src oncoprotein sequences and in several other intracellular signal-transducing proteins such as SOCS2.

## Figures and Tables

**Figure 1 genes-13-01769-f001:**
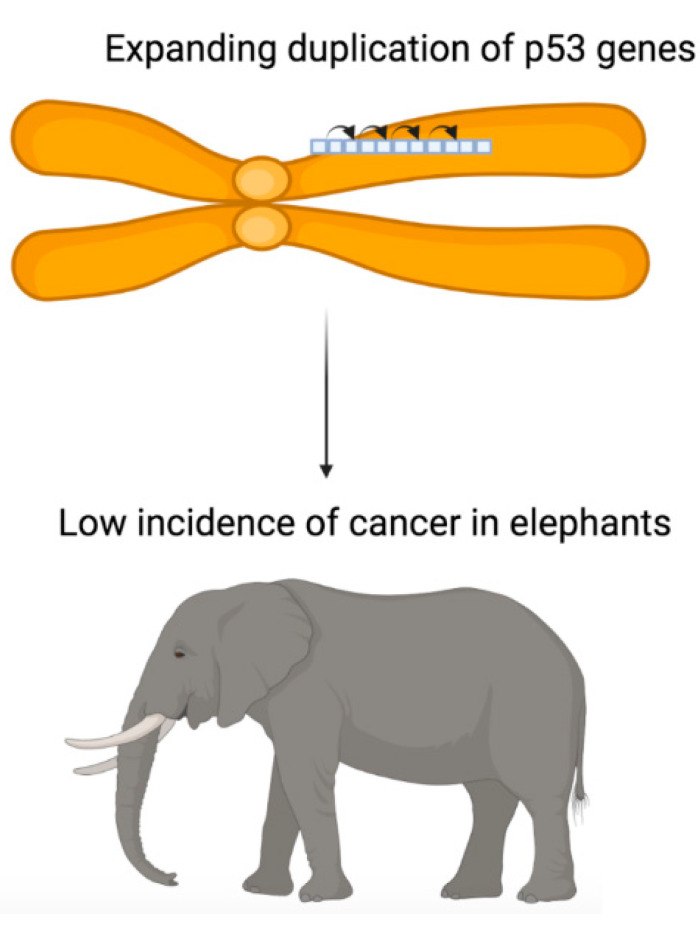
The massive tandem duplication of the p53 gene may promote stronger protection against DNA damage in elephant cells, especially cancers, than in other species such as humans. This figure was created using BioRender (https://biorender.com/, accessed on 1 July 2022).

**Figure 2 genes-13-01769-f002:**
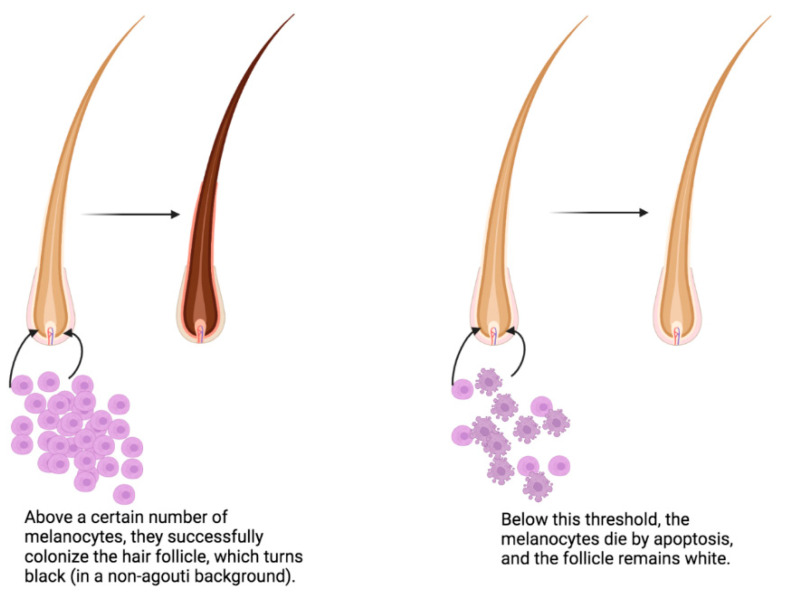
Below a certain number of melanocytes that are able to colonize the hair follicle, they die by apoptosis (right), leading to a white hair. Above this threshold, they survive, leading to a black hair. This results in a mixture of white and black hairs on the coat of the mouse. This figure was created using BioRender (https://biorender.com/, accessed on 1 July 2022).

**Figure 3 genes-13-01769-f003:**
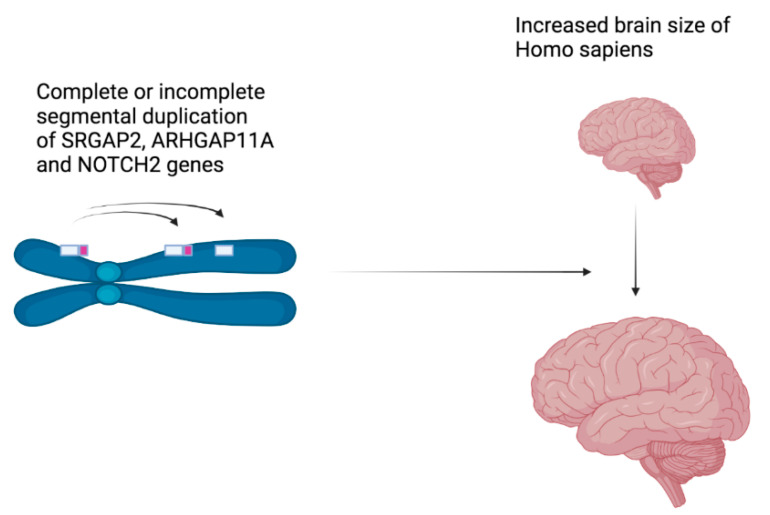
The partial or complete duplications of the SRGAP2, ARHGAP11A and NOTCH2 genes 5 MYA produce proteins that are able to enhance the proliferation of radial glial cells, thus delaying their differentiation to neurons. This figure was created using BioRender (https://biorender.com/, accessed on 1 July 2022).

**Figure 4 genes-13-01769-f004:**
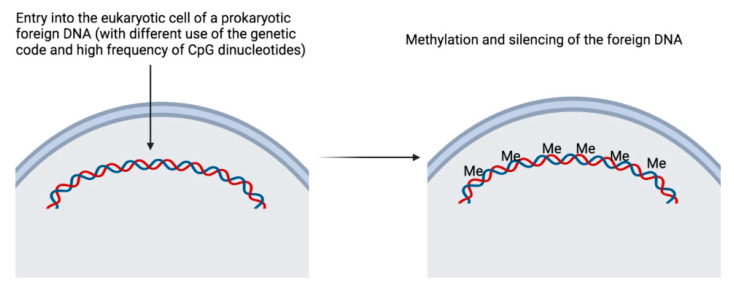
In a mouse oocyte, a foreign DNA (from a species far enough in the tree of life) is recognized as a foreign sequence because it contains an abnormally high number of CpG sequences and the preferential use of the genetic code is not the same. This leads to epigenetic silencing via methylation of the DNA during early development. This figure was created using BioRender (https://biorender.com/, accessed on 1 July 2022).

## Data Availability

Not applicable.

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
