# Peer review of "The Crazy Biology"

_genes, 2022, doi:10.3390/genes13101769_

Round 1

Reviewer 1 Report

The overall essay is crisp and precise; the author has well-drafted the theme in such a way that it is easy to understand by the target audience. The author has explained p53 and its functions and how it is linked with phenotypic characteristics. Furthermore, the author has given a glimpse of all the sub-headings that he has included in this essay and tried to present the fascinating discoveries over a few decades that the research community has crossed. Including a food-based approach (Subheading) can give a positive weightage to the essay.

 Simple and easy to understand, author tried to correlate different aspects with multiple sub-topics in an essay and mentioned the signalling pathways. The author has also considered and thought of comparing elephants to humans. Why do humans have a big brain and a few signalling pathways as well; he also talks about how p53 reduces the risk of cancer in elephants and how massive duplication in elephants than humans. The author has spoken of the analysis part (ChiSquare analysis of codon).

Major points

Varying font size, grammatical errors, and typos were observed throughout the essay. The author didn't give any conclusion, but he raised multiple questions that must be addressed. It is strongly recommended to rework the reference part with reference manager software since it has numerous errors.

Food aspect can be added (how food style regulates p53)

Specific and detailed pathways can be added (try to add figure)

Try to conclude than having questions (conclusion part)

No reference for African elephant(First paragraph)

Author Response

Varying font size, grammatical errors, and typos were observed throughout the essay. The author didn't give any conclusion, but he raised multiple questions that must be addressed. It is strongly recommended to rework the reference part with reference manager software since it has numerous errors.

--> I just have verified and I don't see any reference error....

Food aspect can be added (how food style regulates p53)

--> Not the topic of this essay...

Specific and detailed pathways can be added (try to add figure)

--> I don't see what figure to add.

Try to conclude than having questions (conclusion part)

--> This essay talks about five very different topics, so it is very difficult to write a more relevant conclusion unless the reviewer has a suggestion to make? And in addition to questions that are quite specific to each topic, we have added a general sentence: "Biology complexity still resists and challenges current biotechnologies even the most effective. We still need to work on them, but overall to expand interdisciplinary collaborations to challenge new and exciting biological concepts. For example...."

No reference for African elephant(First paragraph)

--> Ref 2: 

  1. R.D. Sulak, et al.

TP53 copy number expansion is associated with the evolution of increased body size and an enhanced DNA damage response in elephants

eLife 5 (2016), e11994

Reviewer 2 Report

Referee Report

·        Abstract section should be expanded.

·        If possible in-silo analysis should be added.

·        Within the introduction a new important article should be added as reference.

Discovery of sulfadrug-pyrrole conjugates as carbonic anhydrase and acetylcholinesterase inhibitors, Oct 2021, ARCHIV DER PHARMAZIE

·        The resolutions of the figures should be increased.

·        Conclusion section should be revised.

MINOR REVISION

Author Response

Abstract section should be expanded.

--> I have added the following sentence at the beginning of the abstract:"Since the end of the 1980s and the advent of molecular biology, then the beginning of the 2000s with the sequencing of whole genomes, modern tools have never ceased to amaze us and provide answers to questions that we didn't even dare ask ourselves before:"

  • If possible in-silo analysis should be added

--> This essay talks about results in biology, in silico analyzes are tools, not results... I don't see what should be added in this area.

  • Within the introduction a new important article should be added as reference.

Discovery of sulfadrug-pyrrole conjugates as carbonic anhydrase and acetylcholinesterase inhibitors, Oct 2021, ARCHIV DER PHARMAZIE

--> I do not understand what this reference. would do in this essay, the subject matter has nothing to do with it...

  • The resolutions of the figures should be increased.

--> The figures were made with the Biorender tool, which is one of the most powerful tools for drawing figures. I really don't see how I could improve on these figures!

  • Conclusion section should be revised.

--> I have added the following sentence at the end of the conclusion: "I hope that young researchers starting their careers will be able to meet these challenges."

Reviewer 3 Report

Accepted in its present form.

Author Response

Many thanks to the reviewer!